

# An efficient computational framework for gastrointestinal disorder prediction using attention-based transfer learning

Jiajie Zhou, Wei Song, Yeliu Liu and Xiaoming Yuan

Huai'an First People's Hospital, Nanjing Medical University, Jiangsu, China

## ABSTRACT

Diagnosing gastrointestinal (GI) disorders, which affect parts of the digestive system such as the stomach and intestines, can be difficult even for experienced gastroenterologists due to the variety of ways these conditions present. Early diagnosis is critical for successful treatment, but the review process is time-consuming and labor-intensive. Computer-aided diagnostic (CAD) methods provide a solution by automating diagnosis, saving time, reducing workload, and lowering the likelihood of missing critical signs. In recent years, machine learning and deep learning approaches have been used to develop many CAD systems to address this issue. However, existing systems need to be improved for better safety and reliability on larger datasets before they can be used in medical diagnostics. In our study, we developed an effective CAD system for classifying eight types of GI images by combining transfer learning with an attention mechanism. Our experimental results show that ConvNeXt is an effective pre-trained network for feature extraction, and ConvNeXt+Attention (our proposed method) is a robust CAD system that outperforms other cutting-edge approaches. Our proposed method had an area under the receiver operating characteristic curve of 0.9997 and an area under the precision-recall curve of 0.9973, indicating excellent performance. The conclusion regarding the effectiveness of the system was also supported by the values of other evaluation metrics.

## INTRODUCTION

Gastrointestinal (GI) disorders negatively impact various components of the digestive system, such as the stomach, rectum, esophagus, and large and small intestines (*Mayer & Brunnhuber, 2012*). Common among these are acute pancreatitis, intestinal failure, ulcerative colitis, and colorectal cancer. While not all GI disorders are life-threatening, individuals suffering from these conditions often experience significant discomfort in their lives and face increased stress in managing their effects. The 2016 annual report revealed that an estimated 2.8 million individuals received diagnoses of GI disorders, with more than 60% of these cases resulting in fatalities (*Shah et al., 2018*; *Sung et al., 2021*; *Morgan et al., 2022*). Colorectal cancer (CRC), the third most prevalent cancer among men and the second leading cause of cancer-related deaths in women globally, presents a significant health concern (*Kumar et al., 2022*; *Morgan et al., 2022*). Data from

Corresponding author
Jiajie Zhou, hayywcwk2021@163.com

the International Agency for Research on Cancer in 2020 indicated that CRC constituted approximately 10.7% of all new cancer diagnoses (*Araghi et al., 2019*; *Siegel et al., 2020*). GI disorders are increasingly recognized by epidemiologists as a critical challenge for the global healthcare system in the 21st century (*Shah et al., 2018*). Numerous risk factors have been implicated in the etiology of CRC, notably the excessive consumption of fast food, red and processed meats, addictive substances such as alcohol and tobacco, a diet deficient in fruits and vegetables, irregular meal patterns, and a sedentary lifestyle. Reducing these risk factors in combination with regular physical activities helps significantly improve human health (*Garrow & Delegge, 2009*). Besides, periodic and timely diagnosis of GI diseases at an early stage contributes to preventing the growth of cancer (*Mathur et al., 2020*).

To early detect GI disorders and their main causes, gold-standard techniques such as colonoscopy, endoscopy, and biopsy are usually employed due to their broad visual range and convenience (*Peery et al., 2012*; *Wu et al., 2017*). Wireless capsule endoscopy (WCE) is used to inspect the patient's GI tract by moving a capsule (integrated with a wireless, lighted micro-camera) along the tract to examine abnormalities in the mucosal layers. Once swallowed, the capsule smoothly moves down from the patient's throat to the small intestines, transmitting real-time video (*Wang et al., 2013*). The whole process is carefully executed without hurting the patient's throat. The resolution of the video is set at two frames per second to record $256 \times 256$-pixel video frames, facilitating conversion to JPEG-formatted images. An inspection usually takes more than 2 h and may generate approximately 60,000 images for a patient, making a deep analysis of the entire set almost impossible and very time-consuming. After completing the scan, the clinician methodically reviews all the received video frames to detect any abnormalities (*Wang et al., 2019*). Since the abnormal regions may appear in only several frames, the rest of the irrelevant frames are less informative. Despite the fact that all standard hospitals are equipped with modern diagnostic analyzers and skilled technicians to perform the process, the traditional review method is too slow to manage a large number of patients' scanning videos, especially given the ceaseless increase in GI disorders today (*Khanam & Kumar, 2022*). To reduce the workload for technicians and save time and costs, more efficient approaches are required to expedite the process. The review process is also challenging, even for experienced gastroenterologists, because gastrointestinal surgeons may encounter situations that differ from the diagnosed profile. Nevertheless, to improve the chances of successful treatment, early diagnosis is still highly recommended (*Wang et al., 2020*; *Wang et al., 2021*). To address this challenge in diagnosing GI disorders, computer-aided diagnostic (CAD) methods can be used to automate the process and reduce the risk of missing important signs.

In recent years, Artificial Intelligence (AI) applications have been utilized to address complex types of structured and unstructured data. The use of AI for diagnosing diseases based on biomedical images is no longer novel in various medical domains, including binding motif identification (*Zhou & Troyanskaya, 2015*), ADMET assessment (*Kumar, Kini & Rathi, 2021*), investigation of DNA variation (*Alipanahi et al., 2015*), single-cellomics, olfaction (*Sharma, Kumar & Varadwaj, 2021*), and cancer detection (*Pham et al., 2019*; *Jojoa Acosta et al., 2021*). Numerous machine learning (ML) and deep learning (DL)

algorithms have been employed to develop these automated AI-based applications (*Chen et al., 2017*; *Jojoa Acosta et al., 2021*; *Zhang et al., 2021*; *Muthulakshmi & Kavitha, 2022*; *Nguyen et al., 2022*; *Nouman Noor et al., 2023*). The initial success of these computer-aided diagnosis (CAD) methods has motivated scientists to expand their application to various sub-domains, including gastroenterology (*Ruffle, Farmer & Aziz, 2018*; *Kröner et al., 2021*). These computational models actively learn, analyze, and capture distinct characteristics to predict outcomes from any input image. Several automated AI-based applications for the diagnosis of GI disorders have recently been introduced with satisfactory results (*Iakovidis & Koulaouzidis, 2015*; *Fan et al., 2018*; *Sharma et al., 2023*).

*Deeba et al. (2016)* proposed an algorithm to detect percentages of abnormal regions that can be applied to a wide range of diseases, with a specificity of 0.79 and a sensitivity of 0.97. *Yuan, Li & Meng (2017)* classified WCE images of polyps, ulcers, and bleeding using K-means clustering, with an average accuracy of 0.89. *Yuan, Li & Meng (2017)* used support vector machines (SVM) to develop a model for classifying ulcer and non-ulcer regions based on textural features extracted by the contourlet transform and LogGabor filter. *Yuan & Meng (2017)*'s approach achieved an accuracy of 0.94, which is higher than other methods. Another SVM-based model, by *Suman et al. (2017)*, was presented to identify WCE images of ulcers and non-ulcers using multiple color bands in combination with textural features. Their model obtained an accuracy of 0.98. *Souaidi, Abdelouahed & El Ansari (2018)* proposed an SVM-based model for multi-scale diagnosis of ulcers based on extracted texture patterns with an accuracy of 0.94. An SVM-based system, by *Liaqat et al. (2018)*, distinguishes three states of disease, including ulcer, bleeding, and healthy, with an accuracy of 0.98. It is observed that among various ML algorithms, SVM is the most frequently used in addressing this topic.

*Wang et al. (2015)* designed a system for real-time detection of polyps using image flows at a speed of ten frames per second. These images were analyzed using the color texture analysis method. *Korbar et al. (2017)* developed a prediction model for identifying the types of colorectal polyps based on input histopathological images using deep neural networks (DNN) with an accuracy of 0.93. *Komeda et al. (2017)* constructed a model for classifying colorectal polyps using convolutional neural networks (CNN) trained on 1,210 weight-light endoscopy (WLE) images, with an accuracy of 0.75. *Fan et al. (2018)* also proposed using CNN to build a model for detecting ulcers and erosions (in the GI tract) using WCE images as inputs with an accuracy of 0.95. *Sena et al. (2019)* introduced another CNN-based model to predict phases of malignant tissues from histopathology images, with an overall accuracy of over 0.95. *Alaskar et al. (2019)* designed a CNN-based system for ulcer detection to deal with low-contrast and uncommon lesions from endoscopy. *Lai et al. (2021)* implemented a DNN-based model that can capture features of images derived from narrow-band imaging(NBI) and WLE. Their study pointed out that the model trained with full-color NBI images showed better performance compared to one trained with WLE images. Recently, *Sharif et al. (2019)* presented a CNN-based model for classifying GI disorders based on geometric features, with an accuracy of 0.99.

Although existing CAD systems were reported to have high accuracy and sensitivity, they face the risk of being biased in predicting new samples as most of them were developed

using small datasets. Additionally, work reproducibility and slow image-processing time are also issues that need to be solved. Hence, in our study, we propose a more efficient CAD system developed using attention-based Transfer Learning(TL). Four pre-trained networks: ConvNeXt (*Liu et al., 2022*), DenseNet-161 (*Huang et al., 2016*), EfficientNetV2 (*Tan & Le, 2021*), and ResNet-18 (*He et al., 2015*) are used as the main networks for feature extraction, and additional linear layers are placed after the main networks for fine-tuning to learn specific features of images. To boost learning efficiency, the Scale-dot Product attention mechanism is employed to help the model focus on important regions (*Vaswani et al., 2017*).

## MATERIALS AND METHODS

### Benchmark dataset

To develop our method, we used the KVASIR dataset (*Pogorelov et al., 2017*). The dataset contains 8,000 images, divided into three groups with eight sub-groups. The group *Phatological Findings* (disease detected) has three sub-groups: *Esophagitis*, *Polyps*, *Ulcerative Colitis*. The group *Polyp Removal* is specified by two sub-groups: *Dyed and Lifted Polyps* and *Dyed Resection Margins*. The group *Anatomical Landmarks* (normal colon) has three sub-groups: *Z-line*, *Pylorus*, and *Cecum*. Each sub-group comprises 1,000 images whose resolution is $224 \times 224 \times 3$ pixels. We prepared three sets of data: a training set, a validation set, and an independent test set with 7120, 80, and 800 images, respectively. Table 1 gives information on the number of images in the training, validation, and independent test sets.

### Esophagitis

Esophagitis is a medical condition characterized by the inflammation of the esophagus, the tube that connects the throat to the stomach (*Atkins & Furuta, 2010*). This condition can arise from various causes, such as acid reflux, where stomach acids back up into the esophagus, leading to irritation and inflammation. In some cases, esophagitis is triggered by infections, particularly in individuals with weakened immune systems (*Atkins & Furuta, 2010*; *Cianferoni & Spergel, 2015*). Other potential causes include certain medications that can irritate the esophagus lining or allergic reactions to particular foods or substances. Symptoms of esophagitis commonly include difficulty swallowing, chest pain, particularly behind the breastbone, and a sensation of food being stuck in the throat. In severe cases, it may lead to complications like scarring or narrowing of the esophagus. Diagnosis typically involves endoscopy, where a doctor examines the esophagus using a camera, and sometimes includes taking a tissue sample for analysis (*Veerappan et al., 2009*). Treatment varies based on the underlying cause but often involves medications to reduce acid levels, antibiotics for infections, or dietary changes for allergy-related esophagitis (*Rothenberg, 2009*).

### Polyps

Polyps are clustered cells that commonly develop on the inner lining of the colon and sometimes in other parts of the gastrointestinal tract, like the stomach (*Colucci, Yale & Rall, 2003*). They are usually benign, but some types can evolve into cancer over time. The formation of polyps is often linked to genetic factors and lifestyle choices, with diet and age

**Table 1  Data used for model training, validation, and testing.**

| Data | Number of images |
| --- | --- |
| Training | 7,120 |
| Validation | 80 |
| Independent test | 800 |

being significant contributors. These clustered cells vary in size and appearance, and while many do not cause symptoms, larger polyps can lead to abdominal pain, rectal bleeding, or changes in bowel habits. Regular screening, such as colonoscopy, is crucial for early detection, as polyps often present no immediate symptoms (*Fenlon et al., 1999*). Treatment typically involves removal during a colonoscopy, which is a preventive measure against potential malignant transformation (*Bond, 1993*). After removal, polyps are analyzed to determine their type and potential risk. Maintaining a healthy lifestyle, including a balanced diet and regular exercise, is advised to reduce the risk of polyp development.

### Ulcerative colitis

Ulcerative colitis is a chronic condition characterized by the inflammation of the colon and rectum's innermost lining (*Bitton et al., 2001*). This inflammation often results in the formation of ulcers, leading to discomfort and other symptoms. The condition typically manifests in a pattern of recurrent episodes, with periods of intense symptoms followed by times of remission. Common symptoms include abdominal pain, frequent and urgent bowel movements, bloody stool, and fatigue. The exact cause of Ulcerative Colitis remains unclear, but it is believed to involve immune system malfunction, possibly triggered by environmental factors in genetically predisposed individuals (*Roberts-Thomson, Bryant & Costello, 2019*). Managing the condition usually involves a combination of medications aimed at reducing inflammation and controlling symptoms, and in some cases, surgery might be necessary. Lifestyle adjustments, including diet changes and stress management, can also play a supportive role in managing the condition. Due to its chronic nature, patients often require long-term treatment and regular medical follow-ups (*Feuerstein & Cheifetz, 2014*).

### Normal colon

The normal colon, a vital part of the human digestive system, plays a crucial role in the processing and elimination of waste material (*Beynon et al., 1986*). Structurally, it forms the last part of the digestive tract, extending from the small intestine to the rectum. A healthy colon is characterized by its uniform appearance, absence of abnormal growths like polyps, and smooth lining without signs of inflammation or ulceration. Functionally, it is responsible for absorbing water and electrolytes from digested food, thereby aiding in the formation of stools. The normal colon also harbors a diverse microbiome, which is essential for maintaining digestive health, synthesizing certain vitamins, and supporting the immune system (*Chen, Pitmon & Wang, 2017*). Regular screening and colonoscopies are recommended for maintaining colon health, particularly as one ages, to check for any irregularities that might indicate disorders such as CRC or inflammatory bowel disease.

A balanced diet high in fiber, adequate hydration, and regular physical activity are key to maintaining a healthy colon (*Bingham & Cummings, 1989*).

## Transfer learning

Transfer learning (TL) is a technique in the field of ML that involves applying knowledge gained from solving one problem to a different but related problem (*Torrey & Shavlik, 2010*). This approach is particularly useful when there is a scarcity of data available for the target task. Essentially, a pre-trained model, developed for a task with abundant data, is repurposed and fine-tuned to improve performance on the new task. This process not only saves time and resources but also enhances learning efficiency, as the model does not need to learn from scratch. TL has found wide applications across various domains, such as image recognition and natural language processing. By leveraging pre-trained networks, it enables more accurate and efficient training of models on specific, often limited datasets. This approach is particularly beneficial in scenarios where collecting large amounts of labeled data is impractical or too costly. In this study, we used four pre-trained networks, including ConvNeXt (*Liu et al., 2022*), DenseNet-161 (*Huang et al., 2016*), EfficientNetV2 (*Tan & Le, 2021*), and ResNet-18 (*He et al., 2015*), as the main networks for feature extraction. Additional linear layers are added after the main networks for fine-tuning to learn distinct features of images. To enhance learning efficiency, a generic attention mechanism is integrated to help the model focus on essential regions (*Vaswani et al., 2017*).

### ConvNeXt

ConvNeXt, developed by *Liu et al. (2022)*, is a novel CNN-based architecture designed to address the limitations of existing methods. This network can be used to handle a wide range of tasks, demonstrating versatility across various applications. It achieves this by implementing a refined structure that optimizes data processing, making it more efficient than its predecessors. This architecture is characterized by its layered approach, which allows for a more nuanced and detailed analysis of input data as well as facilitates improved learning capabilities, enabling the model to extract and efficiently interpret complex patterns. ConvNeXt has shown significant promise in areas such as image and pattern recognition, where its advanced processing abilities lead to enhanced performance. The development of ConvNeXt marks a significant step forward in the realm of DL, offering a more robust network for tackling complex tasks.

### DenseNet-161

DenseNet-161, a distinctive variant within the DenseNet family, has dense connectivity patterns (*Huang et al., 2016*). This network stands out for its unique approach to connecting each layer to every other layer in a feed-forward manner. Unlike other DeseNet variants, in DenseNet-161, each layer receives concatenated information from all preceding layers, leading to substantial feature reuse and hence a reduction in the total number of parameters. This characteristic makes it a remarkably efficient network in terms of computational resources while maintaining or enhancing performance. The architecture is specifically designed to optimize information flow throughout the network, which helps alleviate

issues like vanishing gradients during training. The depth and complexity of DenseNet-161 may explain its exceptional performance when coping with various challenging tasks, particularly in image classification and recognition.

### EfficientNetV2

EfficientNetV2, an upgraded version of EfficientNet, is known as a robust network that balances speed and accuracy (*Tan & Le, 2021*). This version builds on the foundational principles of its predecessors, focusing on enhancing efficiency. This network is improved to have shorter training speeds and higher accuracy, achieved through refinements in network design and training methods, including the model's depth and resolution of input images, contributing to effective resource utilization and hence reduced computational cost. EfficientNetV2 has shown excellence in various tasks, especially in image classification, where its ability to handle diverse image sizes and complexities offers significant advantages. The development of EfficientNetV2 represents a notable advancement in creating neural networks that are not only powerful but also resource-efficient.

### ResNet-18

ResNet-18, deriving from the ResNet architecture (*He et al., 2015*), is characterized by 18 layers. This network is designed with residual blocks to tackle the problem of vanishing gradients, a common challenge in training DNN. These blocks allow the model to skip one or more layers by employing shortcut connections, thereby preserving the strength of the gradient flow. This innovation not only simplifies the training of deeper networks but also enhances overall performance, making it a good choice for tasks requiring feature extraction and image classification. Its relatively fewer layers, compared to more complex versions in the ResNet family, make it a more efficient and manageable option for various computing environments without significantly compromising on accuracy.

## Attention mechanism

To enhance the focus on specific regions of images in fine-tuned layers, we used Scaled Dot-Product attention, a simplified and commonly used attention mechanism (*Vaswani et al., 2017*). This mechanism operates on the principle of calculating attention by scaling the dot products of Queries and Keys. Essentially, it involves dividing the dot products by the square root of the dimension of the Keys, which helps in stabilizing the gradients during training. This approach enables the model to efficiently weigh the significance of different parts of the input data. By focusing on relevant parts while processing sequences, this attention facilitates a more nuanced understanding and representation of the input. It is especially crucial in tasks that involve processing sequences, like language translation or text summarization, where context and the relationships between different elements in a sequence are key. The computation of Scaled Dot-Product attention is:

$$Attention(Q, K, V) = softmax(\frac{QK^T}{\sqrt{d_k}})V, \tag{1}$$

where $Q$, $K$, $V$ denote *Query*, *Key*, and *Value* vectors, respectively, and $d_k$ is a dimension vector.

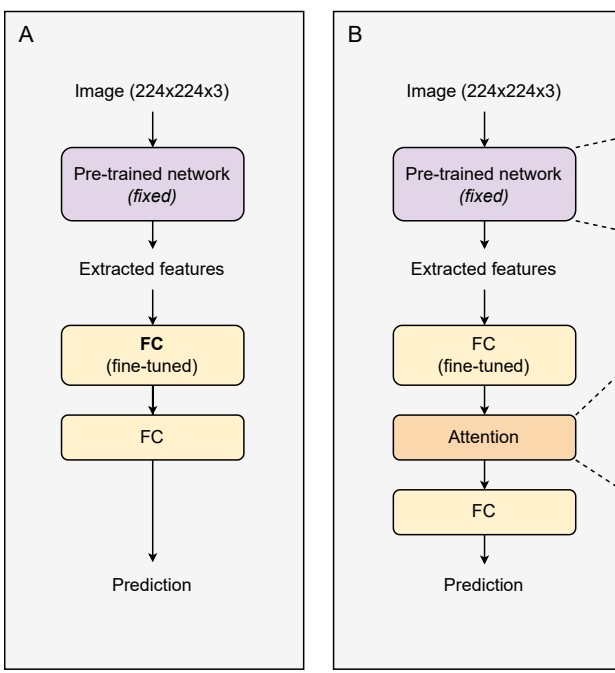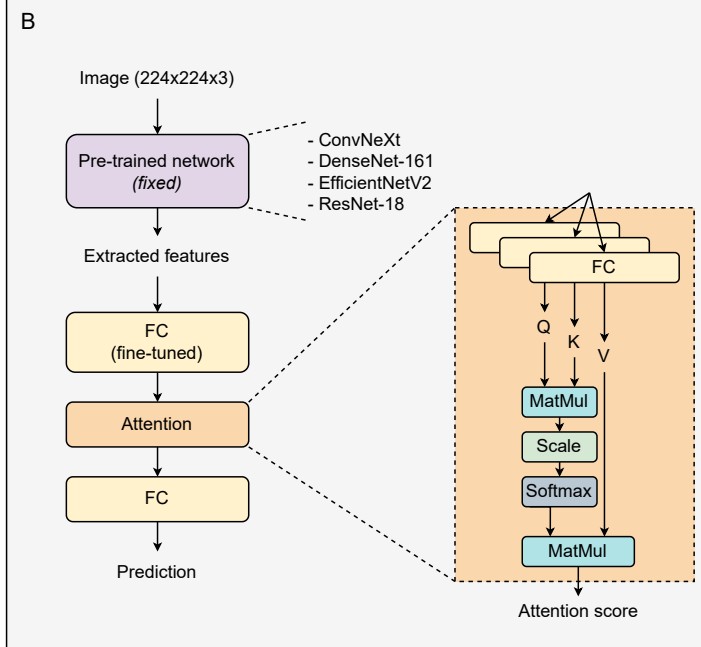

**Figure 1   The proposed architectures of a baseline system and our CAD framework.** (A) Baseline transfer learning system, (B) attention-based transfer learning system.

## Proposed model
### Model training

Figure 1 displays the proposed architecture of the CAD system. Initially, images with a resolution of $224 \times 224 \times 3$ enter a pre-trained network. This network is responsible for extracting features from the images, which are then used in the subsequent layer. Utilizing pre-trained networks for image feature extraction significantly saves time, resources, and effort. Besides enhancing performance, these networks can mitigate the risk of overfitting, especially in scenarios with small datasets, due to their robustness and comprehensive initial training. During training, the weights of the pre-trained network were fixed. The size of the extracted feature vectors varies depending on the type of pre-trained network used. These feature vectors are then fed into a fully connected (FC) layer for fine-tuning before passing through an Attention block. The attention block contains three parallel linear layers for the transformation of feature vectors into corresponding Q, K, and V vectors. The attention output vectors are passed through a linear layer and then activated by the Softmax function to return prediction probabilities. All models were trained over 30 epochs and optimized using the SGD optimizer (*Ruder, 2016*) with a learning rate of 0.001. During the training process, only the linear layers (after the main network) were fine-tuned, while the weights of the main network remained unchanged. The loss function used was cross entropy.

*Metrics for model evaluation*

To evaluate our models, we used various metrics, including the area under the receiver operating characteristic (ROC) curve ($AUC_{ROC}$), area under the precision-recall curve ($AUC_{PR}$), accuracy (ACC), Matthews correlation coefficient (MCC), F1 score (F1), recall (RE) and precision (PR). Except for $AUC_{ROC}$ and $AUC_{PR}$ –threshold independent metrics, other metrics were computed at the default threshold of 0.5. The formulas for these metrics are expressed as:

$$ACC = \frac{TP + TN}{TP + FP + TN + FN}, \tag{2}$$

$$MCC = \frac{TP \times TN - FP \times FN}{\sqrt{(TP + FP)(TP + FN)(TN + FP)(TN + FN)}}, \tag{3}$$

$$F1 = 2 \times \frac{PR \times RE}{PR + RE}, \tag{4}$$

$$RE = \frac{TP}{TP + FN}, \tag{5}$$

$$PR = \frac{TP}{TP + FP}, \tag{6}$$

where TP, FP, TN, and FN stand for the count of true positives, false positives, true negatives, and false negatives, respectively. The metrics mentioned above were used to measure the model's performance for each class. To obtain the overall performance of the model across all classes, we computed weighted metrics, in which the weight of each class is the ratio of the class samples to the total samples. All values presented in Tables 2, 3 and 4 are weighted values for the metrics used.

## RESULTS AND DISCUSSION

### Model evaluation

To find the suitable attention mechanism, we applied three different types of attention, including Dot Product, Additive, and Scaled-Dot Product. After training, these three models were examined using the validation set. The results show that the model employing Scaled-Dot Product attention works more effectively compared to those employing other types of attention (Table 2). Hence, we chose Scaled-Dot Product attention for developing our model.

Table 3 indicates the experimental results on the performance of CAD systems developed across four types of pre-trained networks. The results show that the CAD systems developed with ConvNeXt obtain a higher $AUC_{ROC}$ value, followed by the DenseNet-161-based, ResNet-18-based, and EfficientNetV2-based systems (Fig. 2). Although the DenseNet-161-based CAD system achieves an $AUC_{PR}$ value of over 0.92, higher than other CAD systems,

**Table 2** Performances of our CAD systems developed using three types of attention.

| Attention | Metric | | | | | | |
|---|---|---|---|---|---|---|---|
| | $AUC_{ROC}$ | $AUC_{PR}$ | ACC | MCC | F1 | Recall | Precision |
| Dot Product | 0.9932 | 0.9549 | 0.8888 | 0.8741 | 0.8876 | 0.8888 | 0.8953 |
| Additive | 0.9936 | 0.9588 | 0.8975 | 0.8834 | 0.8973 | 0.8975 | 0.9009 |
| Scaled-Dot Product | 0.9944 | 0.9637 | 0.9138 | 0.9016 | 0.9143 | 0.9138 | 0.9162 |

**Table 3** Performance of CAD systems developed across four types of pre-trained networks.

| Pre-trained network | $AUC_{ROC}$ | $AUC_{PR}$ | ACC | MCC | F1 | Recall | Precision |
|---|---|---|---|---|---|---|---|
| ConvNeXt | 0.9889 | 0.9199 | 0.8663 | 0.8480 | 0.8654 | 0.8663 | 0.8711 |
| DenseNet-161 | 0.9882 | 0.9223 | 0.8563 | 0.8360 | 0.8560 | 0.8563 | 0.8580 |
| EfficientNetV2 | 0.9819 | 0.8879 | 0.8238 | 0.7991 | 0.8235 | 0.8238 | 0.8267 |
| ResNet-18 | 0.9860 | 0.9133 | 0.8388 | 0.8162 | 0.8385 | 0.8388 | 0.8421 |
| ConvNext+Attention | 0.9997 | 0.9973 | 0.9775 | 0.9743 | 0.9775 | 0.9775 | 0.9779 |

**Table 4** Performances of our proposed model over five trials.

| Trial | $AUC_{ROC}$ | $AUC_{PR}$ | ACC | MCC | F1 | Recall | Precision |
|---|---|---|---|---|---|---|---|
| 1 | 0.9997 | 0.9973 | 0.9775 | 0.9743 | 0.9775 | 0.9775 | 0.9779 |
| 2 | 0.9942 | 0.9623 | 0.9125 | 0.9003 | 0.9127 | 0.9125 | 0.9149 |
| 3 | 0.9943 | 0.9627 | 0.8738 | 0.8584 | 0.8700 | 0.8738 | 0.8869 |
| 4 | 0.9948 | 0.9662 | 0.9275 | 0.9173 | 0.9277 | 0.9275 | 0.9286 |
| 5 | 0.9944 | 0.9625 | 0.9150 | 0.9031 | 0.9149 | 0.9150 | 0.9166 |
| Mean | 0.9955 | 0.9702 | 0.9213 | 0.9107 | 0.9206 | 0.9213 | 0.9250 |
| SD | 0.0023 | 0.0152 | 0.0373 | 0.0418 | 0.0385 | 0.0373 | 0.0333 |

it is still lower than the ConvNeXt+Attention-based system, which is enhanced by the attention block. In terms of other metrics, the ConvNeXt-based system is still dominant over the others. However, after being enhanced by the attention mechanism, the CAD system shows superior performance compared to ConvNeXt-based models as well as other systems. The addition of the attention mechanism evidently helps to improve the prediction efficiency.

## Model explainability

To investigate the attention-based learning of our model, we visualized the attention maps of eight classes of samples to highlight specific focusing regions across different classes, emphasizing these regions' importance in distinguishing one class from another. The visualization of these attention maps contributes to understanding the model's decision-making process, providing insights into its attention mechanism. Figure 3 illustrates the attention maps returned by the model for eight classes. These visualized maps show that the model focuses on each class's distinct regions. The attention maps of the normal cecum and z-line classes indicate that their right-handed regions are less focused compared to those of the other classes. The attention maps for the esophagitis, polyps, and ulcerative

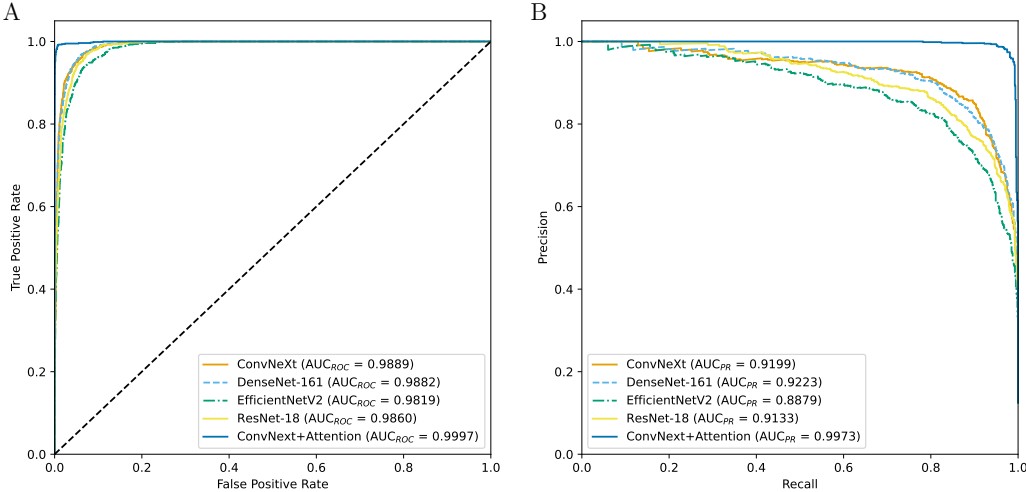

**Figure 2** The ROC curves (A) and PR curves (B) for all the models.

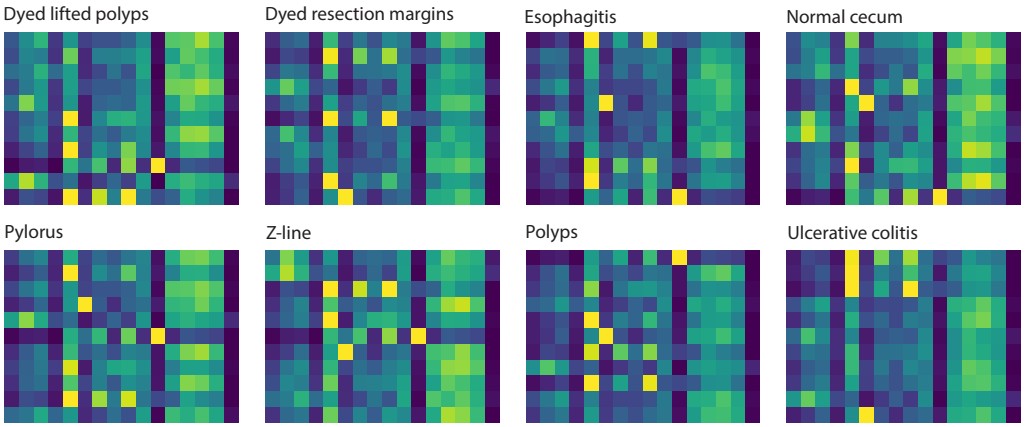

**Figure 3** Attention maps learned by the model correspond to 8 classes for an input sample.

colitis classes reveal that their left-handed regions have darker color bands, which means these regions are more focused than other regions. The darker color bands in the attention maps of disease samples demonstrate that our model effectively learns to distinguish them from normal samples.

In addition to visualizing attention maps, we explored the distribution of samples from eight classes in the training data and test data using $t$-distributed stochastic neighbor embedding ($t$-SNE). The output vectors of the final FC were used as feature vectors for $t$-SNE visualizations (Fig. 4). These $t$-SNE plots show that after passing through the attention layer, all classes are almost separately distributed except for the polyps and dyed lifted polyps classes. The overlapping population between these two classes can be explained by their similar shapes found in some locations. Overall, the model has effectively learned to separate all classes to perform classification tasks.

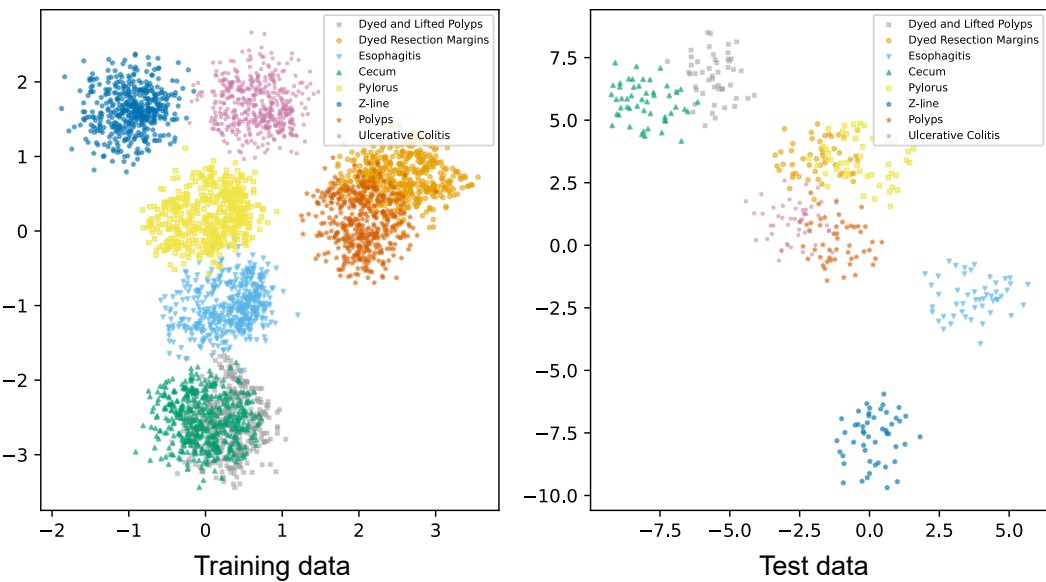

**Figure 4  t-SNE visualizations of samples of eight classes in the training and test data.**

## Stability evaluation

To examine the robustness and stability of the model, we conducted the experiments five times, each with a different random sampling seed for dividing the dataset into the training, validation, and test sets. This approach was designed to evaluate the variation in the model's performance across these distinct partitionings. The results consistently revealed very minor fluctuations in performance metrics, such as accuracy, precision, and recall, across the different trials (Table 4). This consistency confirms that our proposed method is not only effective but also exhibits a high level of stability, making it reliable for further application and suitable for reproduction in various settings. Such robustness is crucial for ensuring that the model can handle diverse and unpredictable real-world data effectively.

## Limitations

ConvNeXt (*Liu et al., 2022*), DenseNet-161 (*Huang et al., 2016*), EfficientNetV2 (*Tan & Le, 2021*), and ResNet-18 (*He et al., 2015*) are prominent DL architectures, each with unique features and varying performance characteristics. ConvNeXt evolves from the traditional CNN design, similar to ResNet-18, but incorporates insights from Transformers (*Vaswani et al., 2017*), leading to improved image classification performance. Its strength lies in its ability to balance the robustness of CNNs with the efficiency of Transformers, though it might not be as extensively tested as more established models. DenseNet-161, on the other hand, stands out with its dense connectivity pattern, where each layer receives input from all preceding layers, significantly reducing the vanishing gradient problem and enhancing feature propagation. However, this can lead to increased model complexity and computational demands. EfficientNetV2, an advanced version of EfficientNet, excels in balancing speed and accuracy, particularly in mobile and real-world applications, due

to its scalable architecture and focus on efficiency. It boasts improved training speed and higher accuracy but may require more careful tuning for specific tasks. ResNet-18, the simplest among these, uses residual connections to enable the training of deeper networks without the gradient vanishing issue, making it efficient for basic tasks but potentially less effective for more complex applications compared to its more advanced counterparts. Each of these models represents a significant approach to neural network design, with pros and cons. Under the scope of this study, although the ConvNeXt+Attention-based CAD system yields higher performance compared to the others, its limitations need to be considered and addressed in further studies.

## CONCLUSION

In our study, we proposed an efficient CAD system developed by combining the ConvNeXt pre-trained network for effectively extracting the image's features with an attention mechanism for enhancing focus on specific regions of images. Experimental results indicated that our proposed method outperformed the state-of-the-art. The introduction of the attention mechanism showed a significant improvement in prediction efficiency compared to those developed with pre-trained networks only. The method can be extended to address other related problems with promising outcomes.

### Funding
The authors received no funding for this work.

### Competing Interests
The authors declare there are no competing interests.

### Author Contributions
- Jiajie Zhou conceived and designed the experiments, performed the experiments, analyzed the data, performed the computation work, prepared figures and/or tables, authored or reviewed drafts of the article, and approved the final draft.
- Wei Song conceived and designed the experiments, performed the experiments, analyzed the data, prepared figures and/or tables, authored or reviewed drafts of the article, and approved the final draft.
- Yeliu Liu conceived and designed the experiments, performed the experiments, analyzed the data, prepared figures and/or tables, authored or reviewed drafts of the article, and approved the final draft.
- Xiaoming Yuan conceived and designed the experiments, performed the experiments, analyzed the data, prepared figures and/or tables, authored or reviewed drafts of the article, and approved the final draft.

### Data Availability
The Python code used in this study is available in the Supplemental Files.

The data is available at Figshare: Zhou, Jiajie (2024). Data for Gastrointestinal Disorder Prediction. figshare. Dataset. https://doi.org/10.6084/m9.figshare.25334257.v1.

## Supplemental Information

Supplemental information for this article can be found online at http://dx.doi.org/10.7717/peerj-cs.2059#supplemental-information.

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
