# Peer review of "An efficient computational framework for gastrointestinal disorder prediction using attention-based transfer learning"

_PeerJ Computer Science, doi:10.7717/peerj-cs.2059_

## Round 0.1 · original submission · Major Revisions

Please address all the comments from the reviewers and revise the manuscript accordingly. Especially, consider repeating the experiments multiple times with different data splits for statistical analysis of the results.

**Language Note:** PeerJ staff have identified that the English language needs to be improved. When you prepare your next revision, please either (i) have a colleague who is proficient in English and familiar with the subject matter review your manuscript, or (ii) contact a professional editing service to review your manuscript. PeerJ can provide language editing services - you can contact us at copyediting@peerj.com for pricing (be sure to provide your manuscript number and title). – PeerJ Staff

Reviewer 1 ·

Basic reporting

The authors proposed a computational framework for predicting gastrointestinal disorders using attention-based transfer learning. The topic is interesting with practical insights into the biomedical domain.

The manuscript was prepared in professional English with good structure. Backgrounds and Literature Review are sufficiently provided. Their findings support the study's objectives. Besides good points, several issues need to be addressed.

(1) How did you prepare your data for model development and testing? Please clarify the sampling strategy in Subsection "Benchmark dataset"

(2) Is there any motivation behind choosing "Scaled Dot-Product" as the attention mechanism? Did you try with some other attention mechanisms? If yes, how did they perform against your chosen one?

(3) Attention maps (returned by the model) need to be visualized to show differences in focused regions of each class.

Experimental design

The work fits the aims and scope of the journal with clearly defined problems. The model shows contributions in the computation biomedical domain. The method is well described with highlighted information for reproduction.

Validity of the findings

According to the journal's criteria, authors must demonstrate the robustness of their work by conducting multiple repetitions (at least five times or ten times (recommended)). Some statistical parameters, such as 95% Confidence Interval (CI) or standard deviation, across these repeated trials, are required. Additionally, the conclusion should be linked to the research questions.

Reviewer 2 ·

Basic reporting

The manuscript seems to be written in a formal English style and is well-organized. The literature review provides a thorough understanding of the subject matter and is sufficient in itself. The research results are presented in an effective manner by the carefully designed tables and figures. Even with these advantages, the manuscript still needs to be significantly improved before it can be considered ready for publication. In particular, there are a few points listed below that require more explanation.

Experimental design

- This problem is related to a multi-class classification issue. Nevertheless, since the different classes are not taken into account in Equations 2–6, I am not sure how the evaluation metrics were computed.
- Is it possible to provide the plots of ROC curves of the models?
- Is it possible to compare the proposed deep learning models with other machine learning algorithms?

Validity of the findings

The authors may want to repeat the experiments with different dataset splits and then present the evaluation metrics' statistics to strengthen the validity of their conclusion.

Additional comments

A revision is needed for this manuscript.

---

## Round 0.2 · accepted · Accept

The authors have addressed all of the reviewers' comments. The two reviewers have also confirmed the manuscript's significant improvement since the last revision. The manuscript is ready for publication.

Reviewer 1 ·

Basic reporting

The manuscript is well-prepared with professional English. All my concerns have been fully addressed.

Experimental design

The experimental design is good. The visualization of model architecture is indicative and add more values to reader's understanding.

Validity of the findings

Based on what authors provided, the validity of the findings is sufficient.

Reviewer 2 ·

Basic reporting

No further comments

Experimental design

Based on my recommendation, the authors have carried out additional experiments and incorporated more visualizations in the revised version. I have no further comments to add.

Validity of the findings

No further comments

Additional comments

The manuscript can be accepted.